# Molecular Investigation of *Eimeria* Species in Broiler Farms in the Province of Vojvodina, Serbia

**DOI:** 10.3390/life13041039

**Published:** 2023-04-18

**Authors:** Marko Pajić, Dalibor Todorović, Slobodan Knežević, Bojana Prunić, Maja Velhner, Dušica Ostojić Andrić, Zoran Stanimirovic

**Affiliations:** 1Scientific Veterinary Institute of “Novi Sad”, 21000 Novi Sad, Serbia; 2Institute for Animal Husbandry, Belgrade-Zemun, 11080 Belgrade, Serbia; 3Faculty of Veterinary Medicine, University of Belgrade, 11000 Belgrade, Serbia

**Keywords:** broilers, coccidiosis, *Eimeria* spp., biosecurity measures, prevalence

## Abstract

Coccidiosis is a significant poultry disease caused by the *Eimeria* species. This study aims to determine the prevalence of *Eimeria* spp. on broiler farms in Vojvodina, along with the identification of parasite species, and assess the implemented biosecurity measures. The study was conducted on 100 broiler chicken farms (28 small-sized; 34 medium-sized; 38 large-sized farms) from June 2018 to December 2021. One pooled sample of faeces was collected from three to six-week-old chickens from each farm, and assessment of biosecurity measures was carried out using a questionnaire. Using the PCR method, DNA of *Eimeria* was found in 59 samples (59%), while 41 samples (41%) were negative. Four species of *Eimeria* were identified, and their prevalence was the following: *E. acervulina* (37%), *E. maxima* (17%), *E. mitis* (25%) and *E. tenella* (48%). A significant difference (*p* < 0.05) was established in the number of oocysts in flocks from small-sized farms compared to medium-sized farms. It was found that regular implementation of disinfection, disinsection and deratisation measures, as well as all the biosecurity measures, can significantly reduce the occurrence of coccidiosis. These results will help to develop better strategies for the control and prevention of coccidiosis on farms.

## 1. Introduction

Coccidiosis is one of the most frequent and significant poultry diseases in the world. It is caused by parasitic protists (protozoa) of the *Eimeria* genus, which parasitises in epithelial cells of the small and large intestines of chickens [1]. Seven *Eimeria* species have long been known to infect chickens and are found on all continents where chickens are bred [2]. The haemorrhagic form of coccidiosis is caused by *E. tenella*, *E. necatrix* and *E. brunetti* [3], while the malabsorptive form of the disease (reduced growth and poor feed conversion rate) is caused by *E. maxima*, *E. acervulina*, *E. mitis* and *E. praecox* [4]. All seven species parasitise in different segments of the intestine. *E. necatrix* and *E. tenella* are considered to be the most pathogenic species [5]. *E. acervulina*, *E. maxima* and *E. tenella* occur most frequently. They are also the most important from economic point of view [6]. In most cases, mixed infections are found [7], usually with two or more *Eimeria* species.

Coccidiosis control measures are implemented through the use of anticoccidial drugs, vaccines and good farm hygiene practices [8,9]. However, the emergence of resistant species of *Eimeria* reduced the effectiveness of anticoccidial drugs. Over time, coccidia has developed resistance to coccidiostats [10]. Resistant *Eimeria* species are the main cause of subclinical coccidiosis (decrease in body weight of chickens, poor feed conversion) and increasing production costs. According to economic data from 2016, the damage caused by coccidiosis on the global level is estimated to be around £10.36 billion [11]. During the last decade, many studies have shown that the use of phytobiotics [12] or their combination with synthetic coccidiostats [13] or ionophores [14] can effectively prevent the occurrence of the disease.

The identification of *Eimeria* species is of considerable importance because it is the basis for effective disease control. The morphological diagnostic method based on microscopic examination of oocysts (oocyst size and shape, sporulation time, prepatent period) was one of the first ways to identify *Eimeria* species [15,16]. However, there are several limitations of this method, due to the similarity and overlapping among oocysts, which has reduced its effectiveness [17]. In addition to microscopic examination, clinical examination of chickens and examination of specific lesions on certain segments of the intestine are used to identify *Eimeria* species [18]. PCR method (polymerase chain reaction) is used as the most accurate method for identification of *Eimeria* species [9].

Due to increasing demand for poultry meat and eggs, industrial poultry production is often faced with a high density on chickens on farms, which increases the possibility of persistence of this parasitosis [10]. Epidemiological research and identification of *Eimeria* species play an important role in prevention and control of coccidiosis. In order to carry out the control effectively, it is important to know which *Eimeria* species are found on farms, together with the risk factors associated with the occurrence and pressure of certain *Eimeria* species. As no epidemiological studies on the prevalence and identification of *Eimeria* species in broiler chickens have been conducted in Serbia, the aim of this study was to determine the prevalence of *Eimeria* and identify the species using molecular methods, and to establish the level of farm biosecurity.

## 2. Materials and Methods

### 2.1. Studying Flocks and Sample Collection

This study was conducted between June 2018 and December 2021 on broiler farms in northern Serbia, the Province of Vojvodina. Faecal samples were collected from a total of 100 broiler flocks and from the same number of farms (Figure 1). Broiler chickens of the “ROSS 308” and “COBB 500” hybrids from small- (<5000 chickens; 28 flocks), medium- (5000–10,000 chickens; 34 flocks) and large-sized farms (>10,000 chickens; 38 flocks) were used in the study. The term “flock” refers to broiler chickens of the same age, health and immune status, which are kept in the same poultry house and make up one epidemiological unit. One pooled sample of faeces from 3- to 6-week-old chickens was taken from each flock. These samples were used for microscopic and molecular diagnosis of *Eimeria* species at the Department of Clinical Bacteriology, Mycology and Parasitology, at Scientific Veterinary Institute “Novi Sad”. All the samples originated from chickens reared on the floor. Faeces were sampled directly from the litter, from 20 to 50 spots inside the facility, mostly in the area of drinking troughs and feeders. During the sampling, the movement path in the facility was in the shape of the letter “W”, according to the method described by Kumar et al. [19]. The samples were taken using sterile gloves and placed in disposable plastic bags and then transported to the laboratory at a temperature of 4 °C.

### 2.2. Processing of Faecal Samples

The faecal samples were transported to the laboratory in a portable refrigerator after collection (WAECO TCX-35, Shenzhen, China). Parasitological examination of faeces was carried out using a modified McMaster flotation method in order to determine the number of coccidia oocysts per sample unit (oocyst per gram of faeces—OPG). Each sample was first homogenised, and 4 g of faeces were separated into a 75 mL plastic bottle (Dunavplast, Inđija, Serbia), and then 56 mL of flotation solution (NaCl, glucose monohydrate) was added. After mixing it for 5 min, 10 mL of the solution was filtered through a strainer into a 10 mL plastic test tube (Spektar, Čačak, Serbia). After 2 min, a part of the solution was transferred from the surface of the test tube using a Pasteur pipette (Boeco, Hamburg, Germany) to the McMaster chamber (Chalex Corporations, Centreville, MD, USA), and 2 to 5 min later the contents of the chambers were examined at 20× and 40× magnification (microscope Olympus BX40F, Tokyo, Japan). Oocysts were counted in each chamber field. The number of oocysts obtained by counting both surfaces of the chamber was multiplied by 50, in order to obtain a total number of oocysts per gram of faeces.

After counting, oocysts were separated for molecular identification, again using a modified flotation method according to McMaster. The sample was transferred from the vial to the test tube, where it remained for 2 min, and 2–3 mL of the solution were transferred from the upper surface to a 50 mL plastic test tube (ProMedia, Kikinda, Serbia) using a Pasteur pipette. After that, distilled water was added to a 50 mL test tube. The content was centrifuged at 750× *g*, and the supernatant was discarded. Then, the sediment with oocysts was transferred to a 2 mL tube where the content was mechanically crushed using a glass rod, in order to destroy the oocyst wall and release deoxyribonucleic acid (DNA).

### 2.3. Methodology for Determination of Subclinical and Clinical Coccidiosis

Subclinical and clinical coccidiosis was assessed based on the findings of microscopic examination of faeces and observed clinical symptoms during farm visits and sample collection. Each flock that was positive on microscopic examination and did not display characteristic clinical symptoms of coccidiosis (bloody or watery diarrhoea, lethargy, lack of appetite, pale combs and wattles, droopy posture and wings, ruffled or puffed-up feathers and poor growth) was marked as a subclinically affected flock. The flocks that were positive on microscopic examination and exhibited clinical symptoms of coccidiosis were marked as clinically affected flocks.

### 2.4. Preparation of Reaction Mixture for PCR

DNA extraction was performed from pre-prepared oocyst isolates with a commercial DNA extraction kit (QIAamp DNA Stool Mini Kit, Qiagen, Germany). A commercial kit (Qiagen, Hilden, Germany) containing 1.5 mM MgCl_2_, 200 μM dNTPs and 2.5 U of HotStart Taq DNA polymerase ready for 1 × PCR buffer was used to prepare the reaction mixture. The components were mixed in a 2 mL plastic microtube (Eppendorf, Hamburg, Germany) in the following volumes per sample: 12.5 μL HotStart Taq Master Mix, 0.25 μL forward primer, 0.25 μL reverse of primers (Table 1) and 7 μL of ddH_2_O. The reaction mixture prepared in this way was resuspended in 0.5 mL microtubes (Eppendorf, Hamburg, Germany), to which 5 μL of extracted DNA per sample was added. One positive and one negative control were used in each PCR reaction.

### 2.5. Molecular Characterisation of Eimeria Species Using PCR Method

After short centrifugation in a microspin centrifuge (Bioscan, Riga, Latvia), the samples were processed in a Thermal cycler TECHNE (Bibby Scientific LTD, Stone, Staffordshire, UK) according to the protocol of Haug et al. [20], although each pair of primers in separate PCR was used (simplex PCR). The protocol included an initial denaturation step at 95 °C for 5 min, followed by 35 cycles of denaturation at 95 °C for 30 s, annealing at 58 °C or 65 °C for 30 s (depending on *Eimeria* species, Table 1) and extension at 72 °C for 1 min. A final prolonged extension step at 72 °C for 3 min finalised the PCR process.

Electrophoresis was performed on a Labnet apparatus (Enduro, Horizontal gel System, Woodbridge, VA, USA) in 1 × TAE buffer (TRIS-EDTA buffer) made from 50× *g* concentrated TAE buffer (Thermo Scientific, Vilnius, Lithuania) at a concentration of 0.5%. A gel (2%) was prepared from agarose (Lonza Verviers, Verviers, Belgium) with the addition of 1 μL of ethidium bromide (Serva, Heidelberg, Germany). Ethidium bromide was also added to 1 × TAE buffer for electrophoresis in the volume of 25 μL per 500 mL of 1 × TAE buffer. GelPilot Loading day dye solution (Qiagen, Hilden, Germany) was used to visualise the PCR products on the gel, while the GelPilot Plus Ladder (Qiagen, Hilden, Germany) marker of 1000 bp was used to determine the size of the fragments. The results were read on a UV transilluminator (UVP, Upland, CA, USA).

### 2.6. Data Collection Using a Questionnaire

During sample collection on the farms, a survey was conducted in order to collect the data on implemented biosecurity measures. The survey was composed of a series of questions related to biosecurity measures, in accordance with the Animal Welfare Act (2009) [21]. The complete questionnaire is shown in Table 2.

### 2.7. Statistical Analysis

Descriptive statistical parameters were used as a method in the statistical analysis of the obtained results of the study. These parameters enabled the description of the obtained experimental results and their interpretation. The following statistical parameters were used: arithmetic mean, standard deviation, standard error, the interval of variation and coefficient of variation. Due to the heterogeneity of the data for the number of oocysts, a log10_(value+1)_ data transformation was performed, and the flocks were tested on the farms, using the Duncan test. The Fisher exact test was used to compare the frequencies of biosecurity measures, while the χ^2^ test was used to compare the frequencies of isolated *Eimeria* by farms. Statistical differences were determined at a significance level of 5%, except for the assessment of the effect of implementation of biosecurity measures on the frequency of coccidia infection, where the significance level was 5%, 1% and 1‰. All the obtained results are shown in tables and graphs. Statistical analysis of the obtained results was performed in GrapfPad Prism version 6.00 (GrapfPad Software, San Diego, CA, USA) and Microsoft Office Excel 2010 (Microsoft Corporation, Redmond, Washington, DC, USA).

## 3. Results

### 3.1. Determination of Eimeria Species

Coccidia oocysts were found in 59 samples (59%), while 41 samples (41%) were negative. The PCR method detected four species of *Eimeria*, namely: *E. acervulina, E. maxima, E. mitis* and *E. tenella* (Figure 2).

Coccidia oocysts were found in 18 faeces samples (64.3%) out of a total of 28 collected samples on small-sized farms. Mixed infection was detected in 12 samples (42.8%). *E. acervulina* was found in 12 samples; *E. maxima* was found in five samples; *E. mitis* was found in six samples; and *E. tenella* was found in 13 faeces samples by molecular detection (Table 3). The comparison of the results of the prevalence of isolated *Eimeria* species showed a significant difference (*p* < 0.05) between the prevalence of *E. acervulina* and *E. maxima* and *E. mitis*. The results were the same after the comparison of the prevalence of *E. tenella* with *E. maxima* and *E. mitis* (Table 3).

On medium-sized farms, coccidia was found in 21 (61.8%) out of a total of 34 samples. Infection with two or more species of coccidia was determined in 15 samples (44.1%). Using the PCR method, *E. acervulina* was found in 13 samples; *E. maxima* was found in six samples; *E. mitis* was found in nine samples; and *E. tenella* was found in 20 samples (Table 4). The prevalence of *E. tenella* was significantly higher (*p* < 0.05) than the prevalence of other isolated species of *Eimeria* (Table 4). Additionally, comparing the prevalence of *E. acervulina* with the prevalence of *E. maxima* and *E. mitis*, a significant difference (*p* < 0.05) was established (Table 4).

The number of positive samples from large-sized farms was 20 (52.6%) out of a total of 38 tested samples. Mixed infection was found in 14 samples (36.8%). Molecular detection identified *E. acervulina* in 12 samples, *E. maxima* in six samples, *E. mitis* in 10 samples and *E. tenella* in 15 samples (Table 5). The prevalence of *E. tenella* was significantly higher (*p* < 0.05) than the prevalence of *E. maxima* (Table 5).

A significant (*p* < 0.05) difference was observed by comparing the results of the number of oocysts per gram of faeces (OPG) of the flocks from small-sized farms in comparison with the flocks from the category of medium-sized farms (Table 6). However, by analysing the results of the number of oocysts of flocks from small-sized farms, no significant (*p* > 0.05) difference was found compared to the large-sized farms (Table 6). Additionally, no significant (*p* > 0.05) difference was identified by comparing the number of oocysts of flocks from medium- and large-sized farms (Table 6).

### 3.2. Prevalence of Subclinical and Clinical Coccidiosis

A total of 100 flocks were examined at all farms included in this study. Coccidia oocysts were found in 59 flocks, while 41 were not infected. Coccidial infection was subclinical in the majority of cases. Despite the presence of oocysts in faeces, there were no clinical symptoms of coccidiosis in 51 flocks (51%) of broilers. Clinical symptoms in broilers were found on eight farms (8%). The clinical symptoms included the following: apathy, ruffled feathers, drooping wings, reduced food and water intake, soiled feathers in the area around the cloaca and blood in the faeces. The prevalence of subclinical and clinical coccidiosis is shown in Figure 3.

The prevalence of isolated *Eimeria* species varied. *E. acervulina* was found in 37% of infected flocks; *E. maxima* was found in 17%; *E. mitis* was found in 25%; and *E. tenella* was found in 48% (Table 7). The number of infected flocks with *E. tenella* was significantly higher (*p* < 0.05) than the number of infected flocks with *E. mitis* and *E. maxima* (Table 7). Additionally, the number of positive flocks with *E. acervulina* was significantly higher (*p* < 0.05) than number of infected flocks with *E. maxima* (Table 7).

The following *Eimeria* species were found on the farms which had a clinical form of coccidiosis: *E. tenella*, *E. acervulina* and *E. maxima*. *E. tenella* was the most commonly found *Eimeria* species.

### 3.3. Data from the Survey on the Implemented Biosecurity Measured on the Farms

Biosecurity measures on all farms were assessed based on the data collected by filling out questionnaires. By analysing the results of the survey using the appropriate statistical method, it was determined that the regular implementation of disinfection, disinsection and deratisation measures (with the significance of *p* < 0.001) and all required biosecurity measures (Table 2) significantly (with the significance of *p* < 0.05) affect reduction of the occurrence of coccidia oocysts in faeces samples and the reduction of the incidence of coccidia infection.

## 4. Discussion

Coccidiosis is one of the most common poultry diseases, which causes huge economic losses in industrial poultry farming worldwide [11]. Epidemiological studies on the prevalence of certain species of *Eimeria* of great importance have occurred [22]. *Eimeria* oocysts are widespread; they are transmitted quickly on farms and are resistant to different types of disinfection [23,24]. In the past, the identification of *Eimeria* was carried out by morphometric analysis of sporulated oocysts [25] and by applying a method for estimating intestinal lesion scores [26]. By applying molecular techniques, the limitations of the morphometric method can be bypassed, and it is not necessary to euthanise chickens for necropsy and estimation of lesion score [7,27]. The PCR technique can identify *Eimeria* species that are present on the farm [10,28], which can help in the prevention and control of the disease. PCR methods are used for *Eimeria* species determination due to the high specificity and sensitivity of this assay. In our study, samples originating from 100 flocks of broiler chickens from 100 farms all from the territory of Vojvodina were analysed. Subclinical coccidiosis was found in 51% of flocks (51/100), while clinical coccidiosis was recorded in 8% of flocks (8/100). Coccidial oocysts were not detected on 41% of farms.

Subclinical coccidiosis on farms causes major economic problems due to the negative impact on growth and feed conversion. Broiler chickens examined in our study did not have clinical signs of coccidiosis on the majority of the farms. Farmers did not report any health issues in the flocks, but the chickens tested positive for coccidia. This means that broilers can have a subclinical form of infection, which is considered to be an important factor in the development of necrotic enteritis in chickens [29,30]. One of the main factors in the occurrence of subclinical coccidiosis is the resistance of *Eimeria* to anticoccidial drugs [10,24], which mostly occurs if coccidiostat programs (“shuttle” and rotation program) are not implemented properly. Shirzad et al. (2011) state that the occurrence of subclinical coccidiosis is in correlation with the age of the flock, so it most commonly occurs in chickens aged 5 to 6 weeks [31]. In our research, subclinical coccidiosis was observed in the chickens aged from 3 to 6 weeks, on 51% of examined farms, while 8% of chicken farms of the same age had a clinically manifested form of the disease. The prevalence of coccidiosis in other countries around the world varies [7,10,16,22,32,33] and ranges from 5.55% [34] to 98.0% [8].

It was found that there are four species of *Eimeria* on broiler farms in Vojvodina: *E. acervulina*, *E. tenella*, *E. maxima* and *E. mitis*. The most frequent species within *Eimeria*-positive samples were *E. tenella* (81.35%) and *E. acervulina* (62.71%) (Table 6). These two species have the highest reproductive potential [35], and in mixed infections, *E. acervulina* can reduce the production of oocysts of *E. brunetti*, *E. maxima*, *E. tenella* and *E. necatrix* [22]. This can be described by the crowding effect, which refers to the availability of intestinal epithelial cells and immunogenicity [35]. Many studies conducted in countries from several regions of the world show that both *E. acervulina* and *E. tenella* are most common on chicken farms [7,8,10,22,27,36,37]. Other *Eimeria* species were identified in most cases in smaller percentage [7,22,34,38,39,40,41], which is also presented in our study.

Infections with several *Eimeria* species are very common on farms [42,43]. Gyorke et al. (2013) describe that in our neighbouring country, Romania, there are mixed infections, usually with two, three or four different types of *Eimeria* [22]. A similar phenomenon was described in Greece, where mixed infections with two or more *Eimeria* species were identified in broiler flocks [7]. The failure of zootechnical control and the low quality of animal management are the main reasons for the occurrence of mixed infections and the decrease in body weight of chickens [43].

The results obtained in our study using molecular and microscopic methods show that coccidiosis can be successfully diagnosed and identified. The main concern is what happens in mixed infections when one species of *Eimeria* is potentially dominant and when the other species is present at a lower level. The PCR technique we used was described by Haug et al. (2007) and is specific to *Eimeria* species. It includes primers for each of the seven species (Table 1) [20]. Some studies show that the lower sensitivity of the conventional simplex PCR as against the real-time qPCR which can consistently detect DNA equivalent to a single sporulated oocyst as against the Agarose gel resolution that required about 10 oocysts [44,45]. One of the ways to obtain more accurate diagnostic results in mixed infections is to combine different molecular techniques. Future studies in our area should include other molecular methods in order to obtain even more precise results.

Although anticoccidials are regularly used for preventive purposes, a high prevalence of *E. tenella* and *E. acervulina* was found on broiler chicken farms in Vojvodina. Subclinical forms of infection can reduce the production performance of chickens and disrupt the function and health of the intestine. This fact suggests that it is necessary to undertake as many activities as possible in order to maintain gut health [36]. On small- and medium-sized farms, the prevalence of coccidiosis was >60%, while on large-sized farms, the percentage of isolated *Eimeria* amounted 52.6%. This can be explained by better hygienic conditions and better management practices on large-sized farms (longer rest periods of the facility, requirement of mandatory shoe changing for farm workers, better disinfection, lower stock density of chickens, etc.).

The implementation of biosecurity measures on broiler farms is one of the essential preconditions for the prevention of coccidiosis [46]. Proper farm management and better organisation of production have a significant role in preventing the occurrence or spread of coccidiosis because oocysts of *Eimeria* are ubiquitous in nature and can be easily transmitted within a farm [47]. Due to the high reproductive potential of *Eimeria*, it is very difficult to ensure coccidia-free ambient conditions, especially on farms with intensive production. On big poultry farms, it is very common that several buildings are found in one backyard, with different age categories of chickens. Usually, employees work in several facilities, which increases the risk of transmitting *Eimeria* through shoes and equipment [48]. By introducing the measure of mandatory clothes and shoe changing, this risk can be reduced [49].

In our study, all implemented biosecurity measures significantly contributed to the reduction of coccidia oocysts in faecal samples and the reduction in the frequency of coccidia infection. The greatest contribution (*p* < 0.001) was recorded for disinfection, disinsection and deratisation measures. Graat et al. (1998) point out the importance of the implementation of hygiene measures on farms as a way to prevent coccidiosis [50]. They claim that it is difficult to prove which factors contribute to the occurrence of coccidiosis, but it is affected by poor hygiene on the farm, the use of the same clothing in several facilities for chicken breeding, the presence of other animals on the farm, feeding and drinking systems that are difficult to wash and clean and workers who work on several farms.

One of the main preconditions for reducing the risk of coccidiosis is improvement of biosecurity measures on farms. Continuous education of farm employees, change of coccidiostats in feed, inclusion of phytobiotics and good farm hygiene should be the main goals of all broiler farms in this region.

## 5. Conclusions

This study shows the prevalence of coccidiosis on broiler farms in Vojvodina. From the initial assumption that potentially all seven pathogenic species of *Eimeria* are present on the field, the actual presence of four species was proved. Their prevalence was the following: 48% for *E. tenella*, 37% for *E. acervulina*, 25% for *E. mitis* and 17% for *E. maxima*. All four types have been found on small-, medium- and large-sized farms. Mixed infections were dominant on all three farm types. A prevalence of more than 60% was confirmed on small- and medium-sized farms, with a lower level of biosecurity measures and poorer management. The results of the questionnaire carried out on all farms show that regular implementation of disinfection, disinsection and pest control measures and all necessary biosecurity measures can significantly reduce the occurrence of coccidia oocysts and decrease the frequency of coccidia infection. These results suggest that molecular identification of *Eimeria* species should be used in the future as it provides an insight into the presence, spread and importance of different *Eimeria* species. Moreover, this will help in coccidiosis control on poultry farms with different biosecurity measures.

## Figures and Tables

**Figure 1 life-13-01039-f001:**
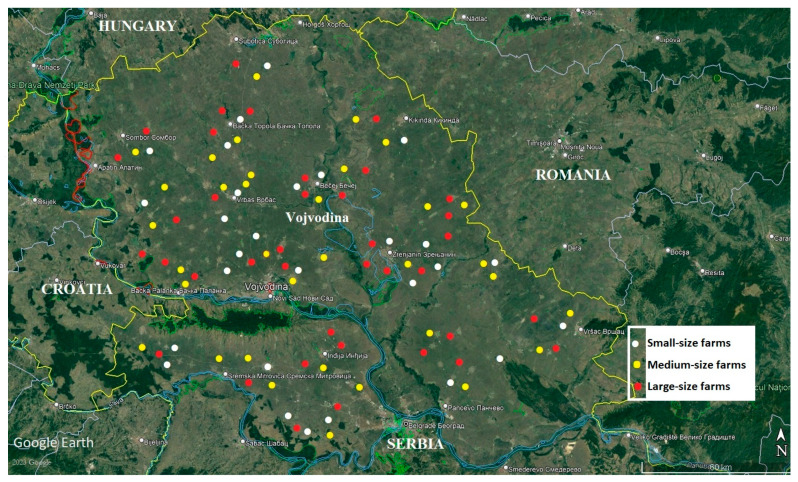
Map of Vojvodina Province and locations of the farms. The red mark shows large-sized farms (>10,000 chickens); the yellow mark shows medium-sized farms (5000–10,000 chickens); and the white mark shows small-sized farms (<5000 chickens).

**Figure 2 life-13-01039-f002:**
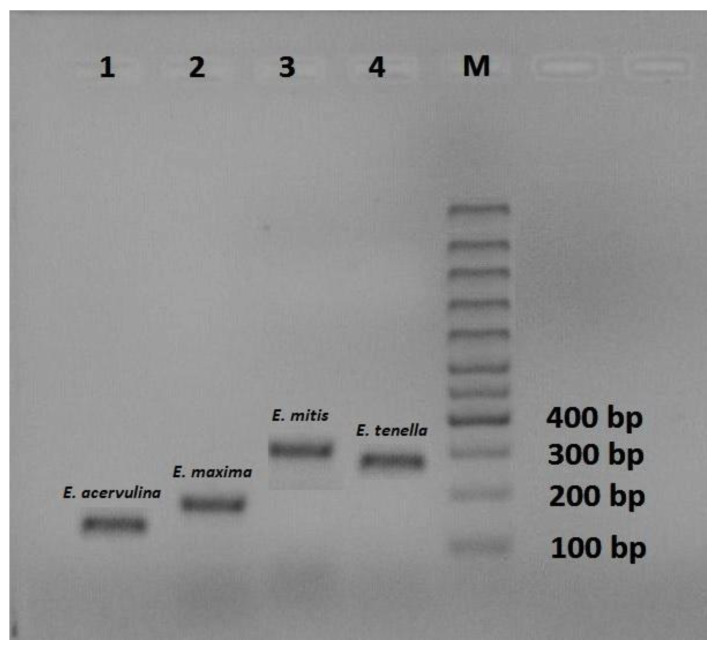
Electrophoresis of PCR fragments of established species of coccidia: 1-*E. acervulina* (145 bp), 2-*E. maxima* (205 bp), 3-*E. mitis* (330 bp) and 4-*E. tenella* (278 bp).

**Figure 3 life-13-01039-f003:**
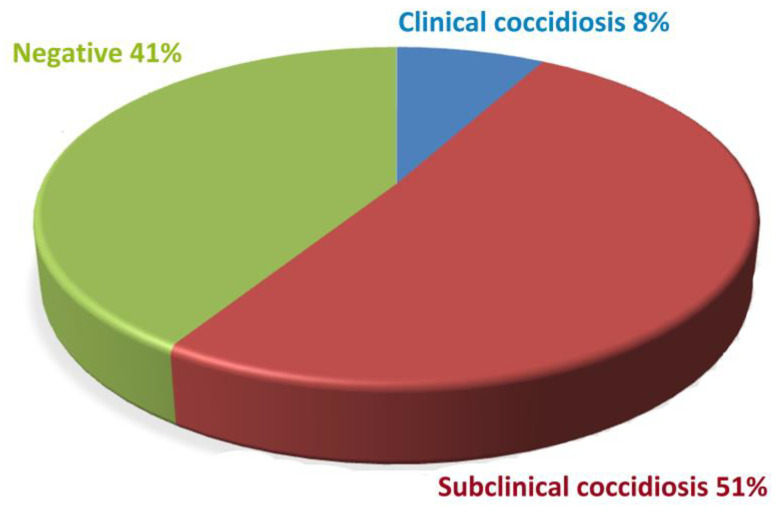
Prevalence of subclinical and clinical coccidiosis on all farm types.

**Table 1 life-13-01039-t001:** List of PCR primers and annealing temperatures used for molecular study (Haug et al., 2007 [20]).

*Eimeria* Species	Primer Name	Primer Sequence	Expected Amplicon Size (bp)	Annealing Temperature (°C)
*E. acervulina*	EAF2 EAR3	5′-GGGCTTGGATGATGTTTGCTG-3′ 5′-GCAATGATGCTTGCACAGTCAGG-3′	145	65
*E. brunetti*	EBF5 EBR3	5′-CTGGGGCTGCAGCGACAGGG-3′ 5′-ATCGATGGCCCCATCCCGCAT-3′	183	58
*E. maxima*	EmaF EmaR	5′-GTGGGACTGTGGTGATGGGG-3′ 5′-ACCAGCATGCGCTCACAACCC-3′	205	65
*E. mitis*	EmiF2 EmiR3	5′-GTTTATTTCCTGTCGTCGTCTCGC-3′ 5′-GTATGCAAGAGAGAATCGGGATTCC-3′	330	65
*E. necatrix*	ENF4 ENR3	5′-AGTATGGGCGTGAGCATGGAG-3′ 5′-GATCAGTCTCATCATAATTCTCGCG-3′	160	58
*E. praecox*	EPF2 EPR3	5′-CATCGGAATGGCTTTTTGAAAGCG-3′ 5′-GCATGCGCTAACAACTCCCCTT-3′	215	65
*E. tenella*	ETF2 ETR	5′-AATTTAGTCCATCGCAACCCTTG-3′ 5′-CGAGCGCTCTGCATACGACA-3′	278	65

**Table 2 life-13-01039-t002:** Questionnaire about biosecurity measures for all investigated farms.

Biosecurity Measures	
Application of the “all in all out” principle?	Is there a vestibule in the poultry house?
Is there a “rest” period for the facility lasting for at least 14 days?	Are disinfection, disinsection and deratisation measures implemented regularly?
Is the farm fenced?	Are carcasses properly disposed of?
Is the farm yard hygiene on a satisfactory level?	Is feeding, drinking and ventilation equipment properly washed and disinfected?
Are there disinfection barriers for vehicles?	
Are there disinfection barriers for people?	Is there a control of visitors who enter the farm?
Do farm workers change their clothes before entering the farm?	Is stocking density more than 15 chickens per square meter?

**Table 3 life-13-01039-t003:** PCR results and number of oocysts in positive samples from small-sized farms.

Farm Type	Positive Flocks								
	Flock Mark	Age (Days)	McMaster Technique	Molecular Identification by PCR				Single Infection	Mixed Infection
			OPG	*E. acervulina*	*E. maxima*	*E. mitis*	*E. tenella*		
Small-sized farms (<5000 chickens)	MS	21	142,100	+	+	−	+	−	+
	CA	27	95,000	−	−	−	+	+	−
	DB	25	74,350	−	−	−	+	+	−
	EV	24	68,750	+	−	+	−	−	+
	CS	30	112,400	+	+	−	−	−	+
	JR	35	121,550	+	−	+	+	−	+
	ZK	30	12,450	+	+	−	+	−	+
	MN	37	28,900	+	−	+	−	−	+
	NN	28	36,250	−	−	−	+	+	−
	PV	27	1750	+	−	−	+	−	+
	SK	28	11,200	−	+	−	−	+	−
	TA	34	24,650	+	−	+	+	−	+
	HI	22	17,700	+	−	+	+	−	+
	PL	24	42,100	−	−	−	+	+	−
	RV	26	22,300	−	+	−	−	+	−
	TI	29	6400	+	−	−	+	−	+
	ZU	32	31,200	+	−	+	+	−	+
	FK	26	29,450	+	−	−	+	−	+
Number of infected flocks Mean ± SD (%)			18/28 (64.3%)	12 ± 0.49 ^a^ (42.8%)	5 ± 0.46 ^b^ (17.8%)	6 ± 0.49 ^b^ (21.4%)	13 ± 0.46 ^a^ (46.4%)	6 (21.4%)	12 (42.8%)

OPG—Oocyst per gram of faeces; Different superscripts show significant differences (*p* < 0.05) in prevalence among different *Eimeria* species.

**Table 4 life-13-01039-t004:** PCR results and number of oocysts in positive samples from medium-sized farms.

Farm Type	Positive Flocks								
	Flock Mark	Age (Days)	McMaster Technique	Molecular Identification by PCR				Single Infection	Mixed Infection
			OPG	*E. acervulina*	*E. maxima*	*E. mitis*	*E. tenella*		
Medium-sized farms (5000–10,000 chickens)	RJ	23	38,500	+	+	−	+	−	+
	LJ	25	13,100	+	−	+	+	−	+
	BD	36	8750	+	−	−	−	+	−
	BR	21	23,200	−	+	+	+	−	+
	CM	37	60,150	−	−	−	+	+	−
	BB	28	12,550	+	−	+	+	−	+
	FG	30	7650	+	+	−	+	−	+
	JN	22	11,300	+	−	+	+	−	+
	KN	38	9350	−	−	−	+	+	−
	LO	21	31,900	+	+	−	+	−	+
	MB	29	14,100	−	−	−	+	+	−
	SB	34	16,950	+	−	+	+	−	+
	SF	33	34,000	+	−	+	+	−	+
	VP	34	26,150	−	+	+	+	−	+
	ML	36	2850	−	−	−	+	+	−
	LN	30	4750	+	−	+	+	−	+
	LM	29	8250	+	−	−	+	−	+
	DL	27	12,000	−	+	−	+	−	+
	CV	27	5450	−	−	−	+	+	−
	NJ	22	29,250	+	−	+	+	−	+
	OM	28	15,350	+	−	−	+	−	+
Number of infected flocks Mean ± SD (%)			21/34 (61.8%)	13 ± 0.50 ^b^ (38.2%)	6 ± 0.46 ^c^ (17.6%)	9 ± 0.51 ^bc^ (26.5%)	20 ± 0.22 ^a^ (58.8%)	6 (17.6%)	15 (44.1%)

OPG—Oocyst per gram of faeces; Different superscripts show significant differences (*p* < 0.05) in prevalence among different *Eimeria* species.

**Table 5 life-13-01039-t005:** PCR results and number of oocysts in positive samples from large-sized farms.

Farm Type	Positive Flocks								
	Flock Mark	Age (Days)	McMaster Technique	Molecular Identification by PCR				Single Infection	Mixed Infection
			OPG	*E. acervulina*	*E. maxima*	*E. mitis*	*E. tenella*		
Large-sized farms (>10,000 chickens)	TF	27	20,550	+	+	−	+	−	+
	PP	25	6150	−	−	−	+	+	−
	VF	25	9600	+	−	+	+	−	+
	OI	29	74,200	+	−	+	−	−	+
	PC	34	41,000	+	+	−	+	−	+
	MZ	33	52,700	−	−	−	+	+	−
	PK	34	40,450	+	+	−	+	−	+
	MP	26	66,650	+	−	+	−	−	+
	FV	22	28,250	−	−	−	+	+	−
	SM	28	49,200	+	−	+	+	−	+
	IA	21	11,050	−	+	+	−	−	+
	DM	26	19,400	+	−	+	+	−	+
	ZV	28	27,350	+	+	−	+	−	+
	MT	32	1950	−	−	−	+	+	−
	VD	35	8700	+	−	+	+	−	+
	VA	29	56,700	−	−	−	+	+	−
	MD	30	20,200	+	−	−	−	+	−
	GK	24	15,050	−	−	+	+	−	+
	GK	25	17,100	−	+	+	−	−	+
	BF	24	24,600	+	−	+	+	−	+
Number of infected flocks Mean ± SD (%)			20/38 (52.6%)	12 ± 0.50 ^ab^ (31.6%)	6 ± 0.47 ^b^ (15.8%)	10 ± 0.51 ^ab^ (26.3%)	15 ± 0.44 ^a^ (39.5%)	6 (15.8%)	14 (36.8%)

OPG—Oocyst per gram of faeces; Different superscripts show significant differences (p < 0.05) in prevalence among different Eimeria species.

**Table 6 life-13-01039-t006:** Faecal oocyst counts (OPG) in flocks from all farm types are shown as the transformed log10_(value+1)_.

Farm Type	OPG Mean ± SD
Small-sized farms	4.49 ± 0.49 ^a^
Medium-sized farms	4.15 ± 0.33 ^b^
Large-sized farms	4.34 ± 0.39 ^ab^

^a, b^ Values within a column with different superscripts are significantly different (*p* < 0.05).

**Table 7 life-13-01039-t007:** Prevalence of different *Eimeria* species isolated from infected broiler flocks from all farm types.

*Eimeria* Species	Number of Infected Flocks Mean ± SD	Prevalence (%)	Prevalence in Relation to the Total Number of Infected Flocks (%)
*E. acervulina*	37 ± 0.49 ^ab^	37.0	62.71
*E. maxima*	17 ± 0.38 ^c^	17.0	28.81
*E. mitis*	25 ± 0.44 ^bc^	25.0	42.37
*E. tenella*	48 ± 0.50 ^a^	48.0	81.35

The different superscripts showed significant differences (*p* < 0.05) in prevalence among different *Eimeria* species.

## Data Availability

Not applicable.

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
