# Peer review of "Molecular Investigation of Eimeria Species in Broiler Farms in the Province of Vojvodina, Serbia"

_life, 2023, doi:10.3390/life13041039_

Round 1

Reviewer 1 Report

The authors present data from a long-term study on the prevalence of Eimeria spp. on farms in Vojvodina, Serbia. Although locally important, the results provide wider, industry- wide implications about biosecurity and sanitation on farm.

There are a few English/grammar points, but no other comments/corrections.

Statistical analysis is appropriate, study design and execution are scientifically sound, and conclusions are appropriate/supported by the data. 

Author Response

Dear reviewer,

Thank you for your time and effort to review our manuscript, and for all your comments and suggestions.

You can find our answers in the attachment.

Marko Pajić,

Reviewer 2 Report

The manuscript authored by Pajic et al., reported the prevalence of coccidiosis in province Vojvodina, Serbia. Overall, the quality of the manuscript is quite low and not suitable to be accepted in Life. The authors investigated for over 3 years and it seems too long to look at the disease prevalence. Also, the study cover very limited region in the specific country which making the manuscript getting low attention to the readers. It is also be suggested for the authors to investigate the infection profile of Eimeria spp. since most of the country have more than 4-5 species of Eimeria infected. It is somehow strange that there are only 4 species in all areas investigated. Please refer the previous reports related. 

Author Response

(The authors gave the same response as above.)

Reviewer 3 Report

Line 17: to identify to identify (delete one)
Line 24: spp. should not be italic
Line 44: It is not a good practice to start sentences with abbreviations 
Line 123: 50 ml should be 50 mL
Line 128: Heading 2.3 is not according to the text. Modify it according to the text.
Heading 2.3: What is the concentration of extracted DNA? How do you measure the DNA concentration?
Line 144: There should be no space between values and signs of a degree centigrade. Check and correct throughout the text.
Methodology regarding the determination of subclinical and clinical coccidiosis is missing.
Statistical values are missing in Tables 3, 4, 5, and 7 and in Figures 3 and 4.
The results of the survey on the applied biosecurity measured on the farms are not sufficient.
Epizootiological data is completely missing in the manuscript; either present the epizootiological data properly or remove this word from the title.
The discussion should be free of results.
Organize the discussion to address each of the experiments or studies for which you presented results.

In the discussion, an explanation of the findings is completely missing

Author Response

(The authors gave the same response as above.)

Round 2

Reviewer 2 Report

The studies focusing on the manuscript are limited.

Author Response

The answers are in the attachment.

Reviewer 3 Report

Lines 194–198: These lines should be in the methodology section.

Figure 2: Labeling of positive samples is missing.

What is new in Figure 3? It should be removed. All the data presented in Figure 3 is already mentioned in Tables 3, 4, and 5.

Author Response

The answers are in the attachment.
